# A hybrid approach to seismic deblending: when physics meets self-supervision

## Abstract

To limit the time, cost, and environmental impact associated with the acquisition of seismic data, in recent decades considerable effort has been put into so-called simultaneous shooting acquisitions, where seismic sources are fired at short time intervals between each other. As a consequence, waves originating from consecutive shots are entangled within the seismic recordings, yielding so-called blended data. For processing and imaging purposes, the data generated by each individual shot must be retrieved. This process, called deblending, is achieved by solving an inverse problem which is heavily underdetermined. Conventional approaches rely on transformations that render the blending noise into burst-like noise, whilst preserving the signal of interest. Compressed sensing type regularization is then applied, where sparsity in some domain is assumed for the signal of interest. The domain of choice depends on the geometry of the acquisition and the properties of seismic data within the chosen domain. In this work, we introduce a new concept that consists of embedding a self-supervised denoising network into the Plug-and-Play (PnP) framework. A novel network is introduced whose design extends the blind-spot network architecture of [27] for partially coherent noise (i.e., correlated in time). The network is trained directly on the noisy input data at each step of the PnP algorithm. By leveraging both the underlying physics of the blending operator and the great denoising capabilities of our blind-spot network, the proposed algorithm is shown to outperform an industry-standard method whilst being comparable in terms of computational cost. Moreover, being independent on the acquisition geometry, our method can be easily applied to both marine and land data without any significant modification.

## 1  Introduction

Reflection seismology [41] is a geophysical technique that uses reflected seismic waves to characterize the Earth's subsurface. It comprises of a controlled source of seismic energy and an array of receivers that record the pressure (or displacement) induced by the reflected waves. After the introduction of 3D seismic [11], today's conventional seismic acquisition campaigns may last several weeks up to a few months [7, 25, 10]. In an attempt to improve acquisition efficiency, and therefore limit the time, cost, and associated environmental impact, [6, 9, 3, 36] introduced a new paradigm in seismic acquisition referred to as *simultaneous shooting*. Simply put, consecutive sources are fired at short time intervals, thereby minimizing the overall acquisition time. This comes at the cost of recording entangled seismic data, also called *blended* data, where the waves originating from one source tend to overlap with those originating from previous and subsequent sources. To render such data suitable for subsequent steps of seismic processing and imaging, the interference between consecutive shots must be removed such that the contribution of each individual source (also referred to as a *shot gather*) is retrieved. This process is called *deblending*. In theory, deblending can be achieved by solving

an inverse problem; however, as this problem is heavily underdetermined, choosing an appropriate regularization is fundamental to achieve a successful inversion. Historically, the design of suitable regularizers is motivated by the effect of the adjoint of the blending operator on the blended data. In fact, the resulting data can be seen as a superposition of coherent signal (i.e, reflections from the shot whose firing time has been properly accounted for) and trace-wise, burst-like noise (i.e, reflections from all other interfering shots whose firing times have not been properly accounted for).

The recent success of deep learning in various scientific disciplines has attracted the interest of the geophysical community, resulting in many opportunities and new challenges [51]. For example, whilst training data should consist of clean, representative ground truth examples that resemble the solution to the inverse problem at hand, such data is generally not available. Two approaches commonly adopted to circumvent this problem are to either generate synthetic data or to use state-of-the-art algorithms to produce input-output pairs to train a network on; in both cases, transfer learning [42, 34] or domain adaptation [2, 12] techniques are then required to generalize the network capabilities to unseen field data. A major drawback of the first approach is that synthetic data may not resemble field data accurately enough to be considered a *representative* dataset: this is well-known in the geophysical community and has been a major criticism for decades when new methods are tested only on synthetic data. It also represents a serious roadblock to the application of deep learning methods in geophysics. Additionally, in most geophysical applications the underlying *physics* is (at least partly) well understood. Pure, end-to-end machine learning methods tend to ignore these well-studied physical principles, thereby discarding important a priori knowledge of the problem they are tasked to solve.

**Our contribution**    We introduce a novel algorithm for seismic deblending, which combines the physics of the underlying physical process with a state-of-the-art self-supervised denoiser into a single, well-crafted inverse process. This is specifically achieved within the framework of Plug-and-Play (PnP) priors. Our network architecture is inspired by the blind-spot network of [27] and modified to handle trace-wise coherent noise. The network is trained on-the-fly at each PnP iteration in a self-supervised manner, completely bypassing the need for ground truth data. Our numerical experiments illustrate that the proposed algorithm can outperform a state-of-the-art conventional method. Finally, we show that our algorithm is independent on the underlying structure of the seismic data and can be used easily for different acquisition set-ups - a clear advantage over conventional methods.

## 2  Background

**The seismic data layout**    Seismic data are commonly acquired by firing a source at a given time and recording the reflections arising from the interaction between the emitted seismic wave and changes in subsurface properties. Conceptually, seismic data can be arranged as a three dimensional tensor (or a cube), having the dimensions of the number of sources $n_s$, number of receivers $n_r$, and number of time samples $n_t$: $d_c(x_s, x_r, t)$. Slicing this cube in different directions gives raise to so-called seismic gathers: more specifically, when slicing across the source axis, we obtain the data recorded by all receivers for a single shot, usually called *Common Shot Gather* (CSG); conversely, by slicing across the receiver axis we obtain the data generated by all shots for a single receiver. When the receivers move alongside the source (i.e., marine case) the resulting gather is called *Common Channel Gather* (CCG). For static receivers (i.e., ocean-bottom or land acquisition), the seismic gather is know as the *Common Receiver Gather* (CRG). Both scenarios will later be considered.

**Blended acquisition**    In practice, to be able to collect data where no overlap exists between consecutive shots, each shot has to be fired with an appropriate time delay, such that all reflections from one shot have been recorded by the receivers before the next shot is fired. This dictates the overall acquisition time and greatly limits any possible acquisition speed-up. Alternatively, in blended acquisition, shots are fired at shorter intervals. This means that each individual CSG contains recordings from both the nominal as well as the previous and subsequent shots. In this work, we consider the so-called *continuous blending* setting. This approach is state-of-the-art in marine seismic acquisition due to the fact it is easy to implement in the field. It is achieved by firing the airgun towed by the acquisition vessel at short time intervals, and continuously recording the waves returning to the receiver array as depicted in figure 1. The recorded data $d_b$ can be simply described as the superposition of all of the unblended, or clean, data shifted in time by the given time delay

$t_i = i \cdot T + \Delta t_i$. Here, $T$ is the nominal firing interval and $\Delta t_i$ is a random dither applied to the nominal firing time of shot $i$. The blended data can thus be described as a function of the clean data

$$d_b = B d_c := [B_1, \ldots, B_{n_s}] \, d_c = B_1 d_{c,1} + \ldots + B_{n_s} d_{c,n_s} \tag{1}$$

where the blending operator is a horizontal stack of time-shift operators $B_i$, and the clean data is a vector where all vectorized shot gathers, $d_{c,i} = vec(d_c(x_{s,i}, x_r, t))$, are stacked together. Moreover, each $B_i$ time-shift operator has the property that $B_i^T B_i = I$, and a composition of time-shift operators is again a time-shift operator [33].

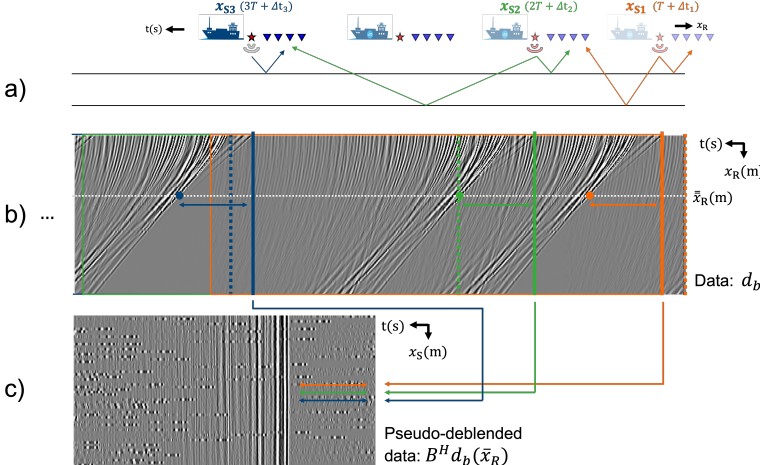

Figure 1: Schematic illustration of a seismic simultaneous shooting acquisition. a) Cartoon of a seismic acquisition campaign in continuous blending mode. A single vessel towing a source (red star) and an array of receivers (blue triangles) moves from right to left and fires energy into the ground at dithered periodic time samples. For each shot, reflections originated from shallow subsurface layers are immediately recorded by the receivers, whilst those produced by deeper reflectors are recorded later in time alongside the shallow reflections from the next firing shot. This phenomenon leads to the blending of independent shot gathers. b) A short time window of the continuously blended seismic data. Dashed vertical color lines represent the nominal firing times (i.e., $i \cdot T$), whilst the solid color lines represent the actual firing times with dithering. Color rectangles refer to every individual shot gather that we wish to separate from the other overlapping gathers. c) Pseudo-deblended data for a single receiver (white dashed line in panel b).

**Pseudo-deblending**  To better understand how to design effective regularization strategies for the deblending problem, we first have to consider the action of $B^H$ on the blended data. For the $i^{th}$ shot gather, the result of $B_i^H B d_c$ can be written as

$$B_i^H \left( B_1 d_{c,1} + \ldots + B_{n_s} d_{c,n_s} \right) = d_{c,i} + \left( B_i^H B_1 d_{c,1} + \ldots + B_i^H B_{n_s} d_{c,n_s} \right) \tag{2}$$

Therefore the action of $B_i^H$ on the blended data produces the original $i^{th}$ shot gather alongside randomly shifted versions of all the other shot gathers. Whilst these randomly shifted shot gathers are coherent and look like standard seismic signal in the CSG domain, they appear as trace-wise coherent noise in the CRG (or CCG) domain as shown in figure 1(c). Because the application of the adjoint of the blending operator retrieves the true signal, albeit with some additional noise, its action is usually called *pseudo-deblending*. As a consequence of this, it is now clear that to retrieve the various $d_{c,i}$, an effective regularization must filter the trace-wise noise in CRGs (or CCGs) whilst preserving the coherent signal. Conventional approaches identify a domain in which the signal can be easily discriminated from the noise, and more specifically the signal in such domain is sparse whilst the noise is not. Examples of such a kind include the hyperbolic Radon transform for CRGs [24], the patched Fourier transform for CCGs [1], or the Curvelet transform [30].

**Deblending by inversion**  Deblending by denoising is achieved by minimizing

$$\min_{d_c} \|d_c - B^H d_b\|_1 + \mathcal{R}(d_c), \tag{3}$$

whereas deblending by inversion amounts to retrieving the clean data by solving the (heavily) underdetermined inverse problem,

$$\min_{d_c} \frac{1}{2}\|Bd_c - d_b\|_2^2 + \mathcal{R}(d_c). \tag{4}$$

where $\mathcal{R}(\cdot)$ is any chosen regularization. The literature has shown that deblending by inversion is superior to deblending by denoising in terms of the overall quality of reconstruction and will be the focus of this work. More details on both approaches are provided in the supplementary material.

**Self-supervised denoising: Incoherent noise**  Self-supervised denoisers are designed in such a way that noisy images can be used as both the input and label to train a neural network to act as a denoiser, thereby bypassing the need for clean data as labels. Noise2Noise represents the first such method not relying on ground truth labels [28]. The network is forced to infer the signal from pairs of noisy data. For applications where such pairs are unavailable, an alternative was proposed in the concurrent works of [26] and [5], who introduced Noise2Void and Noise2Self, respectively. In both cases, the same image is used as input and label: under the assumption that the noise is incoherent whilst the signal is coherent, the network can naturally learn to infer only the signal from its neighbouring pixels. More specifically, to denoise a particular pixel, [26] replace the pixel of the input image with a randomly selected neighbouring pixel. As this pre-processing step introduces randomness in the central pixel of the receptive field of the network, the network should not learn anything from it and naturally learns to infer the signal from its neighbours (since the noise is assumed to be incoherent). Rather than directly replacing the pixel of interest, [5] pre-process the input image with a blind-spot convolutional filter, so that the network cannot rely on the central pixel to predict itself. A key limitation of both approaches lies in the fact that the self-supervised loss can be evaluated only at the pixels that have been corrupted, making the training of these denoisers relatively slow. An alternative approach to blind-spot networks was introduced by [27]. Instead of corrupting the middle pixel, their network is explicitly designed to have a receptive field with a hole in the middle. This is achieved by combining padding and cropping with a standard convolution layer (i.e., to create a causal filter) and by rotating the input image four times prior to feeding it through the network. After the rotated inputs have been fed through the network, they are rotated back, concatenated, and combined by a series of $1 \times 1$ convolutions prior to evaluating the loss at every pixel of the output image. A schematic description of this network is depicted in figure 2a.

**Self-supervised denoising: Coherent noise**  Both Noise2Void and Noise2Self operate under the assumption that the noise is independent and identically distributed. [15] shows that the denoising quality of Noise2Void is degraded when the noise is structured. This shortcoming of Noise2Void is solved by masking pixels along the direction of the noise: the authors dub their method Structured Noise2Void. For the seismic deblending problem, the noise that we are interested to suppress is also structured: more specifically, the blending noise shows correlation along the time axis. We, therefore, extend here the efficient implementation of [27] to suppress structured noise in seismic data, by using the original and flipped (over the source axis) version of the image as input. This produces a network whose receptive field is masked over an entire time trace, see figure 2b. In the following, we will call this network Structured Blind Spot, or StructBS for short.

## 3  Related work

**Simultaneous shooting**  Simultaneous shooting was first pioneered by [6] and has gained popularity in recent years [9]. Although the first attempts at deblending were mostly by means of denoising [33], recent research has reveled the superiority of deblending by inversion [1]. Since then, research has been devoted to finding appropriate regularization terms. Some approaches involve median-filtering [22, 21, 23], rank-reduction methods [17, 55], sparse regularization [29, 30, 56, 57, 59], and deep learning [43, 58, 49]. All the deep learning approaches to date use CNNs and require pre-training. [4] uses the RED framework introduced in [38], which is similar to the PnP framework. The difference is that RED explicitly incorporates the denoiser into the objective function. The authors propose the use of two conventional regularization techniques as a denoiser, the patched Fourier transform [1] and the singular-spectral-analysis filter [17], instead of applying them as a sparse penalty.

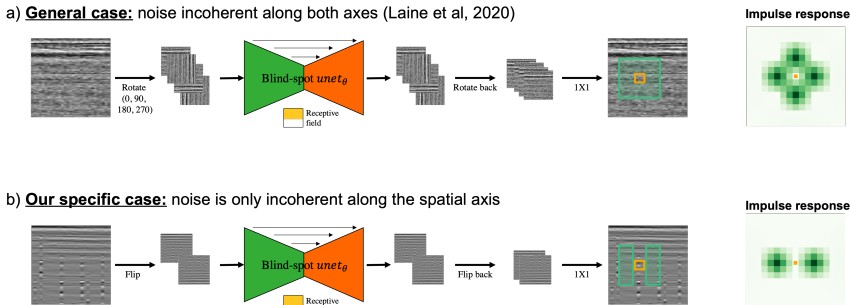

a) **General case:** noise incoherent along both axes (Laine et al, 2020)

b) **Our specific case:** noise is only incoherent along the spatial axis

Figure 2: (a) The blind-spot network of [27], whose receptive field excludes the center pixel. (b) Our newly proposed blind-spot network, whose receptive field excludes an entire direction instead of just the middle pixel. Impulse responses are created by feeding the respective networks with unitary weights and zero biases with an image containing a unitary spike in the middle.

**Self-supervised seismic denoising** Seismic data are a prime example of a noisy data type where no clean, ground truth labels are available. As such, the application of self-supervised denoisers has recently been proposed for the suppression of different types of noise present in seismic data. Following the Noise2Void methodology, [13] use blind-spot networks for the suppression of random noise in post-stack seismic data. Expanding on this, [32] adapted the methodology of StucturedNoise2Void [15] for the suppression of trace-wise noise in seismic shot gathers, originating from poorly coupled receivers and/or dead sensors. The method that is most closely related to ours is the one presented in [49] - both with respect to application and methodology. The authors propose to use a self-supervised denoising network to deblend the data by denoising. To produce satisfactory results they require a number of additional pre- and post-processing steps. In our work, we incorporate a deep learning based denoiser in deblending by inversion, thereby leveraging both the underlying physics and the power of neural networks. Moreover, no pre- and post-processing is required.

**The Plug-and-Play framework** The Plug-and-Play framework was pioneered by [48]. The authors considered a number of popular denoisers, including BM3D [18], K-SVD [19], PLOW [16] and q-GGMRF [45]. In subsequent works, the denoisers have been replaced by pre-trained neural networks, most notably CNN and DnCNN. Lately, [31] proposed regularization by artifact-removal (RARE), a method leveraging a Noise2Noise type approach that requires pre-training. An extensive list of references is provided in [52], and include [38, 54, 35, 47, 20, 46, 29, 44, 53]. This research focuses on progressively training a neural network such that it can adapt to changing noise levels. The novelty of our method is that pre-training is not required.

## 4 Method

Equipped with a self-supervised denoiser, a straightforward approach to deblending is to directly denoise the pseudo-deblended data. However, deblending by denoising is known to be sub-optimal in comparison to deblending by inversion. On the other hand, because a denoiser cannot be naturally added as a constraint to the objective function in equation 4, it is not immediately clear how to incorporate the denoiser into the inversion process. [48] proposed the PnP framework, which is directly derived from the Alternating Direction Method of Multipliers (ADMM). Whilst resembling the alternating minimization process of the classical ADMM algorithm, PnP is more flexible in the sense that it can use any denoiser of choice, without the need for it to be linked to an explicit regularization term for the so-called $y$-update. To understand our method clearly, we give a short derivation of the ADMM following [14], we then link it to the PnP algorithm and finally to our proposed algorithm. The ADMM algorithm is generally used to solve inverse problems of the form

$$\min_x \mathcal{D}(\mathcal{M}(x), d) + \mathcal{R}(x),$$

where $\mathcal{M}$ is the forward model, $d$ is the measured data, $\mathcal{D}$ is a data fidelity term that is generally smooth, and $\mathcal{R}$ is a convex, possibly non-smooth regularization term. Due to the non-smoothness of the objective, this problem cannot be solved with standard gradient-based methods. To account

for the non-smoothness of $\mathcal{R}$, an auxiliary variable $y = x$ is introduced, yielding the equivalent optimization problem

$$\min_{x,y} \mathcal{D}(\mathcal{M}(x), d) + \mathcal{R}(y) \text{ subject to } x = y.$$

ADMM solves this problem by forming the so-called *augmented Lagrangian*,

$$\max_u \min_{x,y} \mathcal{D}(\mathcal{M}(x), d) + \mathcal{R}(y) + \frac{\rho}{2}\|x - y\|_2^2 + u^T(x - y),$$

where $u$ is the Lagrange multiplier and $\rho$ is a scalar. This problem is solved by alternatively minimizing over $x$ and $y$, and maximizing over $u$. This yields the following scheme:

$$
\begin{aligned}
x_{k+1} &= \arg\min_x \left\{ \mathcal{D}(\mathcal{M}(x), d) + \frac{\rho}{2}\|x - y_k + u_k\|_2^2 \right\} \\
y_{k+1} &= \arg\min_y \left\{ \mathcal{R}(y) + \frac{\rho}{2}\|x_{k+1} - y + u_k\|_2^2 \right\} \\
u_{k+1} &= u_k + x_{k+1} - y_{k+1}.
\end{aligned}
$$

The introduction of $y = x$ and the addition of the quadratic penalty $\frac{\rho}{2}\|x - y\|$ yields the $y$-update, which for most popular regularization terms has a simple closed-form solution that can be cheaply evaluated [37]. The key observation of [48] is that the $y$-update can be interpreted as a denoising inverse problem. As such, the authors propose to drop the user-defined regularization $\mathcal{R}(\cdot)$ and instead plug in a denoiser of choice in the $y$-update of the ADMM iterations. Although this may not seem a straightforward choice, PnP has been shown to be competitive (or sometimes even better) than standard regularization methods in a variety of settings. Given the trace-wise structure of the noise and equipped with the self-supervised denoiser, the PnP framework becomes a natural and attractive choice for the deblending task at hand. Our proposed algorithm reads as follows:

$$
\begin{aligned}
x_{k+1} &= \arg\min_x \left\{ \frac{1}{2}\|Bx - d_b\|_2^2 + \frac{\rho}{2}\|x - y_k + u_k\|_2^2 \right\} \\
y_{k+1} &= \text{StructBS}_\theta(x_{k+1} + u_k) \\
u_{k+1} &= u_k + x_{k+1} - y_{k+1}.
\end{aligned}
$$

where $x$ is used here for simplicity in place of $d_c$, and the $x$-update is performed using an iterative solver of choice, e.g. LSQR. The $y$-update is now the denoiser $\text{StructBS}_\theta$, where $\theta$ denote the network parameters. The variable $u$ couples both $x$ and $y$ and forces them to be close together. The $x$-update requires the solution to satisfy the physics dictated by the equation $Bx = d_b$, and the $y$-update denoises the noisy receiver gathers.

## 5 Experiments

In the following, our algorithm is tested on the openly available Mobil AVO viking graben line 12 marine dataset [1]. As the data has been originally acquired in a conventional fashion, we create the blending operator and blend the data ourselves. In addition to containing all the challenging features of a field dataset, this also provides us with a ground truth, $d_c$, onto which to assess the quality of our reconstruction. In this example, the original dataset is composed of $n_s = 64$ sources, $n_r = 120$ receivers, and $n_t = 1024$ samples (i.e., the total recording time per shot equals 4 seconds). For the continuous blending operator, we choose a fixed firing interval of $T = 2$ seconds, with added random delays selected uniformly in the interval $\Delta t_i \sim [-1, 1]$ seconds. This overlap is quite challenging as generally half of the signal overlaps with either that of the previous or that of the next shot. Moreover, some pseudo-deblended shot gathers exhibit contributions from three consecutive shots. Finally, the relative mean-square error, $RMSE = \|d_c - d_{c,true}\|_2/\|d_{c,true}\|_2$, is chosen as a metric of comparison in all of our numerical examples. All experiments are performed on a Intel(R) Xeon(R) CPU @ 2.10GHz equipped with a single NVIDIA GEForce RTX 3090 GPU.

### 5.1 Comparison with state-of-the-art deblending

To begin with, our newly proposed methodology is compared with the state-of-the-art deblending algorithm of [1] that solves the deblending problem as a sparsity promoting inversion

$$z_\star = \arg\min_z \|BFz - d_b\|_2^2 + \lambda\|z\|_1, \quad d_b = Fz_\star, \tag{5}$$

---

[1] https://wiki.seg.org/wiki/Mobil_AVO_viking_graben_line_12

where $F$ is a linear operator that performs a patched two-dimensional Fourier transform, and $z_\star$ is the solution in the Fourier domain that is ultimately transformed back to the original time-space domain of the seismic data. In our experiment, the size and number of patches as well as the regularization parameter $\lambda$ are selected by hand to yield optimal results. Moreover, the FISTA [8] solver is used with an adaptive decreasing sequence, $\lambda_k$ (as this has been shown in the literature to outperform a fixed $\lambda$ for this specific problem). We choose the sequence $\lambda_k = \left(\frac{6}{5}e^{-0.05k} + 6\right)\lambda_0$, which was once again fine-tuned to give the best performance. The final error is roughly $9.8\%$ (see supplementary material for details); hereon in this represents the benchmark against which we will assess the effectiveness of our self-supervised PnP algorithm. Next, our PnP algorithm is applied to the same dataset. We choose 30 outer iterations, 3 inner iterations, $\rho = 1$ and 30 denoiser epochs. The choice of these hyperparemeters will be justified in the ablation study. We also use the U-Net architecture in [27], the $L_1$ norm for the self-supervised training loss because it is more appropriate for burst-like noise, and the Adam optimizer with default parameters. Since the denoiser is trained on all the CCGs, the size of our training data is 120 and we use a batch size of 8. This leads to a solution that has an overall error of roughly $6.7\%$, which is approximately $3\%$ lower than the conventional method. As a visual comparison, figure 3 displays the results for a given CCG (top) and CSG (bottom) for both the conventional and proposed approaches. It is noteworthy that our algorithm shows a clear improvement in terms of denoising capabilities, as visible in the displayed CCG. Especially after $t = 2s$, where the signal is weak and blending noise dominates, the conventional approach tends to be more prone to signal leakage compared to our PnP algorithm. Finally, the computational cost of the conventional algorithm can be quantified in terms of the number of forward and adjoint operations for both the blending ($B$) and patched Fourier ($F$) operators: in our example, this amounts to 200 forward and adjoint passes. On the other hand, our method requires 90 forward and adjoint passes for the blending operator and a total of 900 training epochs for the network. Considering that all computations (apart from the network related ones) are performed on the CPU, the two algorithms are comparable in terms of overall computational time (2h and 34mins for the conventional algorithm and 1h and 51mins for the PnP algorithm).

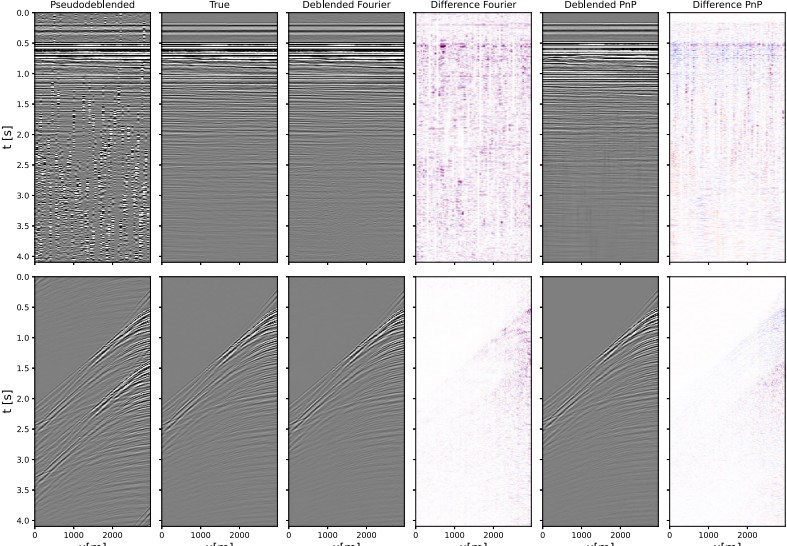

Figure 3: Deblending results for one CCG (top) and CSG (bottom). Although both algorithms can successfully remove most of the blending noise, our algorithm is less prone to signal leakage and provides better amplitude fidelity - a key factor in seismic data processing.

## 5.2 Ablation study

This section provides an extensive analysis of some of the key components of the proposed PnP methodology and their impact on the overall solution of the deblending inverse problem.

**PnP iterations** To begin with, we assess the importance of the PnP iterations compared to simply training the self-supervised denoiser on pseudo-deblended data and applying it directly to the entire dataset. Although not shown here, the result of this *one-shot denoising* produces a solution with an overall error of roughly 19%. This is much worse than both the conventional and PnP method and therefore considered not suitable.

**The $x$-update** The ablation study with regard to the $x$-update is provided in the supplementary material. This includes a study on the effect of the number of inner iterations and the $\rho$ parameter. For our continuous deblending problem, we have shown that they could be safely fixed to 3 and 1. Future experiments with different datasets and blending strategies are required to verify this assumption.

**The $y$-update** In our implementation we propose to start with a randomly initialized network and train it for a fixed number of epochs at every outer iteration. A warm start strategy is employed such that the weights of the network at a given outer iteration are initialized to those obtained at the end of the training of the previous outer iteration. The efficacy of this approach is shown in figure 4a, where we compare on-the-fly training with and without warm starts, where the latter re-initializes the network at every $y$-update. From the error curves, we can safely conclude that warm starting the network is clearly beneficial. Since there is no theoretical justification for this particular strategy, we consider a few other alternative strategies. The first strategy is to use a pre-trained network. Here pre-training is achieved by denoising the pseudo-deblended data in a self-supervised manner; this approach could greatly reduce the computational cost of the overall algorithm since we do not need to train the network at every iteration. A comparison of the relative error with that of the proposed, on-the-fly training shown in figure 4b reveals that after a few outer iterations, the network is unable to further remove the remaining noise in the data.

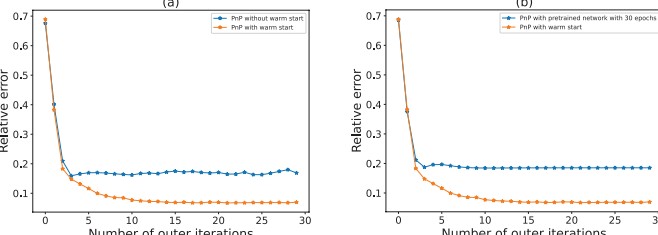

Figure 4: a) Error for network training with and without warm starts. b) Error when running the PnP algorithm with a network pre-trained on the pseudo-deblended data.

Another option is to stop training the network after a few outer iterations. Ideally, the network will have learnt how to remove the noise encountered during the first iterations, and extra training will not improve the denoiser capabilities. We run experiments where we stop the training after a fixed number of outer iterations to see whether there is an added benefit to continuing training the network. Results are shown in figure 5a. In all of the scenarios we clearly see that stopping the training after

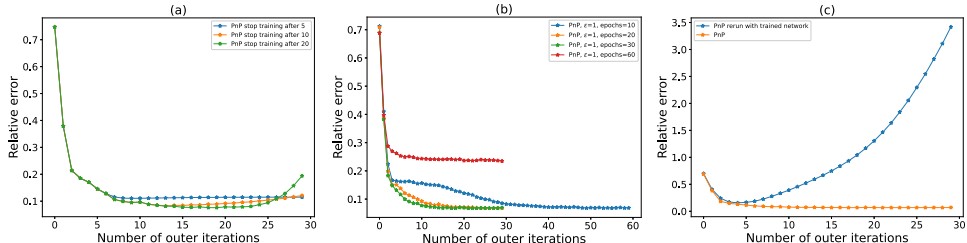

Figure 5: a) Error for on-the-fly training where training is stopped after a certain number of outer iterations. b) Error for different number of training epochs. c) Error when using the network at the end of the PnP algorithm for the entire process versus our proposed on-the-fly training strategy.

a certain number of outer iterations leads to a stagnation in the error, or even worse to an increase

in the error at later iterations. This behaviour is known as *semiconvergence* in the inverse problems community. Both results are perhaps not surprising, as the input to the network at every iteration contains a different noise level compared to that of earlier iterations: the noise in $x_k$ constantly reduces during the overall inversion. Therefore, the network is required to learn a slightly different task at each time. In figure 5b, we assess the impact of the number training epochs for the denoiser. We clearly observe that at some point the curves for 30 and 20 epochs start to coincide, meaning that there is no additional gain in the extra 10 epochs of training. The curve for training with 10 epochs seems stagnant after the first three iterations, but eventually it picks up momentum and goes down again. Note that, in terms of overall epochs, the cost of performing 30 outer iterations with 10 epochs each is the same as using 10 outer iterations with 30 epochs each. However, every outer iteration carries an additional cost of three inner iterations for the $x$-update, which is not negligible as it requires evaluating the forward and adjoint of the blending operator. In general, it seems beneficial to perform more epochs in the early outer iterations, although there is a limit after which the error starts to stagnate. Moreover, training with 60 epochs leads to overfitting.

Finally, to further investigate the generalization capabilities of the network for blending problems, the network weights are saved after the last outer iteration of the PnP algorithm. The PnP algorithm is then re-run using the saved network without performing any on-the-fly training. Figure 5c shows that this strategy fails, illustrating that the network may have forgotten how to deal with the higher noise levels encountered in the early iterations. This results highlights the importance of using a self-supervised denoiser that can be easily and cheaply trained on-the-fly. The use of pre-trained denoising networks such as DnCNN may instead require training multiple networks with different noise levels, unless a bias-free, non-blind network is used [52].

# 6 Limitations and Conclusions

**Limitations** Our algorithm requires the setting of a number of hyperparameters, namely the number of inner and outer iterations, the parameter $\rho$, and the number of epochs for the self-supervised denoiser. The number of epochs seems to have a major impact on the quality of the deblending process and the overall convergence properties of our algorithm. Additional hyperparameters that have not been explored in this work are associated with the network itself, e.g. the number of layers, the activation function, batch size, etc. This is also a direction for further research. Another drawback is that there is no convergence guarantee, since our operator $B$ is underdetermined and therefore not strongly convex [39]. Empirically, we observe that $x_k$ and $y_k$ tend to converge to similar values for some carefully selected hyperparameters, indicating that at least in our experiments the algorithm converges successfully. Similarly, to obtain convergence guarantees for the PnP method, the denoiser has to be Lipschitz continuous; when a neural network is used, this means that spectral normalization is required during training. In [50], it was shown that PnP algorithms can be convergent when combined with carefully pre-trained denoisers that satisfy such condition.

**Societal impact** Blended acquisition greatly reduces the time required to acquire seismic data, thereby limiting the impact of seismic acquisitions on the environment. Apart from shooting at shorter intervals, there is no difference compared to conventional acquisition. Moreover, recent research has suggested that the energy emitted by each source could be lowered. This may provide acquisition solutions that are more environmentally friendly for marine life.

**Conclusions** We have introduced a novel hybrid algorithm for seismic deblending, combining the physics of the blending operator with a self-supervised denoiser that is naturally embedded into the Plug-and-Play framework. We have adapted the network architecture in [27] to enforce an extended blind spot along an entire axis (time, in our case) instead of single pixels. Because the denoiser is self-supervised, our approaches bypasses the need for ground truth labels that are usually unavailable for seismic applications. Experiments on a field dataset have shown that the proposed method can outperform a state-of-the-art, sparsity-based algorithm. Moreover, as show in the supplementary material, our algorithm is independent on the type of acquisition, which is usually an issue for conventional algorithms. Although our algorithm requires the setting of a number of hyperparameters, we have argued that the number of inner iterations and $\rho$ can most likely be set to a fixed number and this easily generalizes to different seismic acquisitions. However, the network architecture and the number of epochs may require tuning for different acquisition setups. We hope to address these issues by having an adaptive strategy for setting the number epochs in future work.

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
