# OpenReview forum: "A hybrid approach to seismic deblending: when physics meets self-supervision"
_NeurIPS.cc/2022/Conference — NeurIPS 2022 Submitted_

### Official Review · Reviewer_okRD · 2022-07-11

**Rating:** 4
**Confidence:** 4
**Soundness:** 2 fair
**Presentation:** 2 fair
**Contribution:** 2 fair

**Summary:**

This paper proposed a plug-and-play (PnP) algorithm for reflection seismology (RS). The key concept of PnP is to leverage an image denoiser as an implicit regularizer to impose prior within an iterative optimization algorithm. By using deep image denoisers, PnP combines the physical constraints and trainable priors. Proposed algorithm in this paper combines PnP with self-supervised deep image denoisers to solve the RS problem.

**Questions:**

1. Besides the apparent benefits of self-supervised deep image denoiser, that is, no use of ground-truth data, what else motivates you to use such image denoiser? Why this is denoiser is necessary.

2. Apart from the deep image denoiser, what else can be considered as contributions of the work?

**Limitations:**

Not applicable.

**Strengths And Weaknesses:**

**Strength:**
- New application of PnP to reflection seismology (RS).

**Weakness:**
- The proposed method is not novel. The claimed novelty is the inclusion of self-supervised image denoiser in the PnP framework. However, this has been done in [1] for medical imaging.
- The presentation of the work is not well organized. The introduction and background is too long while the method section is too short.

[1] RARE: Image Reconstruction using Deep Priors Learned without Ground Truth

---

> ### Author Response · Authors · 2022-08-02
> **Reply to your review**
>
> Dear reviewer, thank you for taking the time to read our paper and your feedback. Please find below our point-by-point response to the issues you have raised.
>
> We would first like to respond to one of the weaknesses you claim, namely that our method is not novel. You cite reference [1], but this work is not the same as ours. Rather, in their work, the authors use the fact that different angles in the measurement setup yields different artifacts for the same underlying clean image. This is a Noise2Noise type of approach, and cannot be always applied to any inverse problem (for example, this is not suitable for the deblending problem). On the other hand, we are required to use a Noise2Void type of approach. Moreover, their work never leverages the structure of the underlying noise, and they apply a standard DnCNN for the denoising. All in all, different scientific problems call for different solutions, and the fact we both use the PnP algorithm and a neural network based denoiser does not make the two methods the same.
>
> 1. “Besides the apparent benefits of self-supervised deep image denoiser, that is, no use of ground-truth data, what else motivates you to use such image denoiser? Why this is denoiser is necessary.”
>
> In lines 168-173 we write:
>
> “For the seismic blending problem, the noise that we are interested to suppress is also structured: more specifically, the blending noise shows correlation along the time axis. We, therefore, extend here the efficient implementation of [28] to suppress structured noise in seismic data, by simply using the original and flipped (over the time axis) version of the image as input. This produces a network whose receptive field is masked over an entire time trace, see figure 2b. In the following, we will call this network StructBS.”
>
> Note, the early work of [28] focused on random noise and required noise to be independent between pixels. When structure existed within the noise field the network would learn to predict both the signal and the structured (correlated) components of the noise field. By adapting the network design, we have ensured the noise correlated along the time axis, i.e. noise on an individual trace, cannot be included in the prediction of the trace’s signal values.
>
> 2. “Apart from the deep image denoiser, what else can be considered as contributions of the work?”
>
> The contribution of our work is not necessarily the denoiser, but rather coupling the denoiser with the underlying physics of the blending problem via the PnP framework. Within deep learning for geophysics, typically physics has either been excluded beyond the training data or the problem has been set up to only solve the known physical equations governing geophysics (i.e., via PINNs). In this work, we have shown for the first time how we can leverage the strength of a self-supervised network in a challenging inverse problem, whilst still ensuring the physics of the underlying problem is satisfied. We believe this is the main contribution, along with the adaptation of the network so that the network is blind to the entire trace.

---

> > ### Comment · Reviewer_okRD · 2022-08-08
> > **Further discussion**
> >
> > Thank you for the response. I am convinced about the first question, but I still have some reservations about the second question. The claim *"In this work, we have shown for the first time how we can leverage the strength of a self-supervised network in a challenging inverse problem, whilst still ensuring the physics of the underlying problem is satisfied"* is overly too strong. RARE, in my opinion, is such a method that combines physics consistency and deep self-supervised priors (meaning no ground-truth/fully-sampled data is needed). By substituting Noise2Void for other deep denoisers in PnP, one cannot consider this method completely new.

---

> > > ### Author Response · Authors · 2022-08-09
> > > **Reply to your response**
> > >
> > > Thank you for taking the time to respond to our reply. The claim may be a bit strong, but it is nowhere in the manuscript, just in our response to your review. We do however believe that our proposed methodology may apply to a larger class of problems than RARE. The Noise2Noise approach requires two samples with the same underlying signal but a different noise distribution, whereas the blind-spot network only assumes either uncorrelated noise or correlation in a certain direction. Moreover, our network is trained on-the-fly instead of a-priori like in the RARE framework, which allows us to automatically adapt to the specific dataset at hand, thereby preventing generalization issues. Based on the above we still feel our work has more differences than similarities with RARE and therefore novel from a standpoint of using deep learning to aid the solution of inverse problems.

---

### Official Review · Reviewer_fDvL · 2022-07-11

**Rating:** 4
**Confidence:** 3
**Soundness:** 3 good
**Presentation:** 2 fair
**Contribution:** 2 fair

**Summary:**

This paper combines a self-supervised deep image denoising method and PnP framework to improve seismic deblending. The problem is interesting and significant. However, the rationale of the proposed algorithm is unclear. It seems like a combination of existing techniques and the technical contribution seems limited.

**Questions:**

1. How is the blending operator decided? Why does it have the property of B^TB=I? Maybe a proper citation is needed here for the reader to better understand the background.
2. What do t(s), x_R(m), and \bar{\bar{x}}_R(m) mean in Figure 1 b)? Also, why does the y-axis x_s(m) instead of x_R(m) in Figure 1 c)? What does m represent here?
4. According to Figure 1 b), the input seems like an image. However, it is confusing how it is generated. What are the white curves and the horizontal dash line mean?
5. What is the regularization term in the proposed optimization problem? How is y-update derived as StructBS(x+u)？
7. How will the model perform if only adopting PnP framework but does not use the self-supervised denoiser?


**Limitations:**

The authors have adequately addressed the limitations and social impact of their work.

**Strengths And Weaknesses:**

Strength:
1. This paper aims to improve the performance of seismic deblending, which is a significant problem and can benefit geophysical studies.
2. The authors explore to improve the conventional deblending problem with deep image denoising technology.
3. The proposed method can outperform a conventional method according to the experiment.

Weakness:
1. This paper is poorly organized and the meanings of many symbols in formulas and figures are not well introduced (see Questions for detail). This makes it difficult to understand the background and the problem formulation. The reader needs to spend a long time reading this paper before understanding the problem.
2. The authors introduced their algorithm very briefly in “Method”. The motivation of the proposed algorithm is unclear because the authors only provide the resulting update rules instead of showing the original optimization problem and the regularization term. Especially, the biggest modification to the algorithm in this paper seems to be using a self-supervised denoiser for y-update. However, the rationale behind it is confusing. I wonder how this update rule is derived. Indeed, too much essential information has been omitted.
3. Partly because the authors did not give any theoretical illustrations for their algorithm, the technical contribution seems limited. According to the paper, the authors (1) follow the conventional optimization problem formulation of seismic blending/denoising; (2) use PnP framework to solve this optimization problem; (3) change a term for y-update (without properly explaining the rationale) based on the self-supervised image denoiser. Then, this paper seems like a combination of existing techniques. The inspiration it can bring to the community is limited.
4. The experiments are not sound. The authors have mentioned in the related work that there are other seismic deblending/denoising methods. However, only one conventional method is used as the baseline. The authors should report their performance to better demonstrate the effectiveness of the proposed algorithm in comparison to SOTA. Otherwise, the authors should explain why the other methods are not applicable.

---

> ### Author Response · Authors · 2022-08-02
> **Reply to your review**
>
> Dear reviewer, thank you for taking the time to read our paper and your feedback. Please find below our point-by-point response to the issues you have raised.
>
> 1. “How is the blending operator decided? Why does it have the property of B^TB=I? Maybe a proper citation is needed here for the reader to better understand the background.”
>
> In lines 97-104 we write:
>
> “The recorded data db can be simply described as the superposition of all of the unblended, or clean, data shifted in time by the given time delay $t_i$ = i · T + ∆ti. Here, T is the nominal firing interval and ∆ti is a random dither applied to the nominal firing time of each shot. The blended data can thus be described as a function of the clean data
>               $d_b = Bd_c := [B_1,...,B_{ns}]d_c =B_1d_c,1 +...+B_{ns}d_{c,ns}$
> where the blending operator is a horizontal stack of time-shift operators Bi, and the clean data is a vector where all vectorized shot gathers, $d_{c,i} = vec(d_c(x_s,i,x_r,t))$, are stacked together. Note that each $B_i$ time-shift operator has the property that $B_i^TB_i = I$, and a composition of time-shift operators is again a time-shift operator.“
> This explains both the blending operator and why it is self-adjoint. In the revised manuscript, we have however added a reference to a seminal paper in geophysics on deblending so that the interested reader can find more details than those we can provide here due to page-length limit.
>
> 2. “What do t(s), x_R(m), and \bar{\bar{x}}_R(m) mean in Figure 1 b)? Also, why does the y-axis x_s(m) instead of x_R(m) in Figure 1 c)? What does m represent here?”
> t(s) stands for time in seconds, x(m) stands for the spatial axis in meters and the subscript denotes either source (S) or receiver (R). We have elaborated on this in revised the manuscript.
>
> 3. “According to Figure 1 b), the input seems like an image. However, it is confusing how it is generated. What are the white curves and the horizontal dash line mean?”
>
> We are not exactly sure what you mean by white curves, but we believe that you mean the actual seismic data. Note that seismic data is generated by waves passing through the subsurface and are therefore typically characterized in seismic data by bands of white-black-white or black-white-black curves which correspond to the actual pressure changes at the recording device. The horizontal dashed line is the trace that corresponds to the pseudo-deblended gather in figure 1c as explained in the caption.
>
> 4. “What is the regularization term in the proposed optimization problem? How is y-update derived as StructBS(x+u)?”
> We agree that we have not stated this clearly enough. We propose to move the derivation of the ADMM from the supplementary material to the main manuscript and shorten the introduction. We will also explain this in more detail. The main point is that there is no explicit regularization term in the PnP framework, this is in fact the key clever idea behind PnP. Rather, in the y-update of the ADMM algorithm, the minimization problem amounts to a denoising step. The entire PnP framework is based on this interpretation, and rather than denoising with a chosen regularization, one can instead apply any denoiser that works well with the data at hand even though it does not necessarily have a quantifiable underlying regularization term (e.g., BM3D, non-local means, K-SVD or neural networks). We will stress this point in the methodology section to make it clearer.
>
> 5. “How will the model perform if only adopting PnP framework but does not use the self-supervised denoiser?”
>
> In this work we have decided to use this specific self-supervided denoiser for 2 main reasons: 1) it seems perfect for the type of noise that arises in this application: trace-wise independent noise; 2) it does not require any training data other than the blended data itself.
> Of course, one could use a different denoiser, like a pre-trained denoising neural network with natural images or a conventional denoiser like BM3D. Both would not make much sense as these are more general denoisers and we do not expect them to perform as well as our denoiser for this type of noise we wish to remove, given the fact that the noise is structured.
> Another option could be to denoise with, for example, the patched Fourier transform used in our other example. It is well known that the patched Fourier as a pure denoiser does not work well, so similarly we would not expect it to work well as the PnP denoiser. Only when it is applied as a preconditioner in the inversion does the patched Fourier method work.
> We do however agree that the point raised here is an interesting point, which we believe it may be worth of a separate study.

---

### Official Review · Reviewer_Vpu2 · 2022-07-11

**Rating:** 4
**Confidence:** 2
**Soundness:** 3 good
**Presentation:** 1 poor
**Contribution:** 1 poor

**Summary:**

The paper introduces the problem of seismic deblending and provide a state-of-the art result in this problem using denoising network in the PnP setup. Denoising model is learned as a part of PnP.

**Questions:**

1. Please consider weaknesses above as questions.

**Limitations:**

Limitations are adequately addressed.

**Strengths And Weaknesses:**

[strengths]
1. Introduces Seismic Deblending to ML community. Seismic problems are generally less explored in ML community and this paper introduces a good problem.
2. Good results with learned denoiser.

[weaknesses]
1. Writing : The paper devotes very less space to the main algorithm. PnP + denoiser. Also, for people who are not familiar with the ADMM algorithm like me, the section 4 reads very difficult with a lot of terms introduced but not explained like y_k , x-update, y-update, LSQR, StuctBS, theta. These details are provided partially in the supplementary. However more discussion space should be provided to the algorithmic part.

2. Novelty: I am not sure if paper provides enough novelty for this conference. It uses denoiser architecture in a seismic setting and shows good results. However, it is not clear what the contributions are apart from applying existing denoiser in seismic PnP.

3.  Experimental validation is limited. The experiments are performed on a single dataset. More empirical validation is needed.

---

> ### Author Response · Authors · 2022-08-02
> **Reply to your review**
>
> Dear reviewer, thank you for taking the time to read our paper and your feedback. Please find below our point-by-point response to the issues you have raised.
>
> 1. “Writing: The paper devotes very less space to the main algorithm. PnP + denoiser. Also, for people who are not familiar with the ADMM algorithm like me, the section 4 reads very difficult with a lot of terms introduced but not explained like y_k , x-update, y-update, LSQR, StuctBS, theta. These details are provided partially in the supplementary. However more discussion space should be provided to the algorithmic part.”
>
> Upon review of the manuscript, we agree with this point. We propose to move the derivation of the ADMM from the supplementary material to the main manuscript and elaborate on the introduction of the denoiser. We hope that this will clear up a lot of your confusion. We have also gone through the paper again with a fine-toothed comb and made sure in our revised manuscript that all terms are defined.
>
> 2. “Novelty: I am not sure if paper provides enough novelty for this conference. It uses denoiser architecture in a seismic setting and shows good results. However, it is not clear what the contributions are apart from applying existing denoiser in seismic PnP.”
>
> We have introduced an algorithm that naturally combines both physics and machine learning, without the need for access to ground truth data. The novelty is that the network is self-supervised and trained on-the-fly, and that pre-training of any sort is not necessary. To the best of our knowledge, people have proposed the use of pre-trained denoisers in the PnP framework, but none of them have suggested to use a self-supervised denoiser. This is a novel contribution that may be valuable in other scientific applications beyond geophysics, where it is not possible to access ground truth input-output data for supervised learning or not even a set of models belonging to the same manifold of the expected solution to train a classical denoiser on to be later used in the PnP framework. Also, the fact that we train a network whilst we solve the inverse problem of interest is something we do not believe exists in the literature: this is a form of continual/transfer learning where the network can naturally adapt to changing noise levels without having to be trained from scratch (and our Figure 4 clearly shows this).
>
> 3. “Experimental validation is limited. The experiments are performed on a single dataset. More empirical validation is needed.”
>
> We understand the concern about the experiment being carried out on a single dataset. However, in geophysics, the quality of the datasets is far more important than the number of datasets the algorithm is applied to. Synthetic datasets lack important features that are present in real data, and therefore the consensus is that applying an algorithm to field data is far more valuable. In our case, the data is synthetically blended, where the blending operator is a time shift that can be easily modeled synthetically. Moreover, we now have a ground truth. Field datasets are often not openly available in the geophysics community, because they are far too valuable for companies. Mobil AVO is one of the few open seismic datasets, and therefore we have chosen to use it in our experiments. Please note that we have done experiments on a different acquisition geometry of the data, which could be regarded as a different dataset.

---

### Official Review · Reviewer_TN2y · 2022-07-13

**Rating:** 6
**Confidence:** 4
**Soundness:** 4 excellent
**Presentation:** 4 excellent
**Contribution:** 2 fair

**Summary:**

This paper tackles the inverse problem of "seismic deblending" by using the well known "Plug-And-Play" (PnP) framework together with a recently proposed self-supervised denoiser, which is adapted to follow the noise structure of the particular problem at hand. Experiments on a real dataset artificially blended by the authors show that the approach outperforms a state-of-the-art method based on a Fourier sparsity prior, and an ablation study show that the proposed combination of PnP with a carefully trained self-supervised denoiser is efficient.

**Questions:**

The main question I have is: Is the proposed method without the PnP steps the same as [47] ? If so, this should be made very explicit in the method section and in the experimental results.

I noted the following minor typos:
- L136 : "to to"
- L172 a image -> an image

**Limitations:**

The limitations and societal impact of the work are adequately addressed by the authors in section 6.

**Strengths And Weaknesses:**

## Strengths
- The paper is generally very didactic. Quite unusually, 5 out of 9 pages are devoted to the introduction, background and related work sections. In that sense, the paper can almost be regarded as a "tutorial" paper, giving a subsequent overview of the field of seismic deblending and of recently proposed self-supervised denoising techniques to non-expert readers.
- The proposed approach is sound and seem to outperform a more conventional technique on a real dataset

## Weaknesses
- While 5 pages are devoted to introductory materials, only half a page is devoted to presenting the methodology. On the one hand this seems short, on the other hand, this can be explained by the fact that the proposed methodological contribution is very incremental: two relatively well-known existing techniques are straight-forwardly combined and applied to a known problem in the field.
- The experiments are relatively limited, eventhough this may be justified by the lack of available real data for this problem. Still, more detailed quantitative results could have been provided.
- No detail is given on the key "StructBS" step of the method (the self-supervised denoising), not the paper nor in the supplementary material. The reader is entirely referred to the recent reference [47] for this (in fact, the reference to [47] is implicit, and the acronym StructBS is never made explicit). This makes the methodological part of paper not self-contained, which is not good for research reproducibility and transmission.
- A lot of important implementation details are scattered across the experimental section or in the appendix, which would make the proposed method very hard to reimplement
- In the ablation study, not using the PnP iterations and using only self-supervised denoising seem to be equivalent to what is done in [47], but this is not explicitly stated by the authors. More details should be given on these results than a single number, since according to the authors themselves, [47] is the closest work to what they propose.

---

> ### Author Response · Authors · 2022-08-02
> **Reply to review**
>
> Dear reviewer, thank you for taking the time to read our paper and your feedback. Please find below our point-by-point response to the issues you have raised.
>
> 1. “While 5 pages are devoted to introductory materials, only half a page is devoted to presenting the methodology. On the one hand this seems short, on the other hand, this can be explained by the fact that the proposed methodological contribution is very incremental: two relatively well-known existing techniques are straight-forwardly combined and applied to a known problem in the field.”
>
> We agree that the introduction is long compared to the methodology section. Given the criticism given by yourself and other reviewers, we propose to move the ADMM derivation from the supplementary material to the methodology section and elaborate on the rationale behind introducing a denoiser in the y-update. Accordingly, we will shorten the introduction.
>
>
> 2. “The experiments are relatively limited, eventhough this may be justified by the lack of available real data for this problem. Still, more detailed quantitative results could have been provided.”
>
> See general comments for the first sentence. Regarding the second sentence, we have provided panels of the deblending error (Figure 3) and RMSE values, this is the common way of reporting deblending results.
>
> 3. “No detail is given on the key "StructBS" step of the method (the self-supervised denoising), not the paper nor in the supplementary material. The reader is entirely referred to the recent reference [47] for this (in fact, the reference to [47] is implicit, and the acronym StructBS is never made explicit). This makes the methodological part of paper not self-contained, which is not good for research reproducibility and transmission.”
>
> The StructBS is explained in lines 168-172:
>
> “We, therefore, extend here the efficient implementation of [28] to suppress structured noise in seismic data, by simply using the original and flipped (over the time axis) version of the image as input. This produces  a network whose receptive field is masked over an entire time trace, see figure 2b. In the following, we will call this network StructBS.”
>
> We also refer to figure 2b, which demonstrates the network design. However, you are correct in that we never explicitly defined the acronym, which stands for Structured Blind Spot. We will include this definition in the revised manuscript.
>
> Also we would like to note that the relevant reference is [28] not [47], as mentioned in your comment. Please see our comments in the general section and below regarding reference [47] and their slightly similar study, which however does not include the physical aspects of the deblending task and as such requires substantial pre- and post-processing steps.
>
> Finally, to your concern about reproducibility, we have provided our entire codebase and reproducible examples in the Supplementary material.
>
> 4. “In the ablation study, not using the PnP iterations and using only self-supervised denoising seem to be equivalent to what is done in [47], but this is not explicitly stated by the authors. More details should be given on these results than a single number, since according to the authors themselves, [47] is the closest work to what they propose.”
>
> This is not equivalent to what is done in [47]. In fact, in lines 191-195 we explain:
>
> “The method that is most closely related to ours is the one presented in [47] - both with respect to application and methodology. The authors propose to use a self-supervised denoising network to deblend the data by denoising. To produce satisfactory results they require a number of additional pre- and post-processing steps. In our work, we incorporate a deep learning based denoiser in deblending by inversion, thereby leveraging both the underlying physics and the  power of neural networks. Moreover, no pre- and post-processing is required.”
>
> Their pre- and postprocessing steps make the method different from pure denoising via deep learning. For example, in their post-processing step (called tuning stage by the authors), the network is kept fixed and optimization is performed directly on what we call the d_b vector: apart from being an expensive choice, as it requires back-propagation all the way to the input of the network, this tuning stage has no theoretical meaning and guarantee. We choose a different approach, and leverage the PnP framework to incorporate the physics. This is an important step, as it is well-known that deblending by inversion is far superior to deblending by denoising: our experiment confirms this.

---

### Author Response · Authors · 2022-08-02
**Reply to all reviewers**

We would like to thank the reviewers for carefully reading our manuscript and for their feedback. We address a few points that have been raised by multiple reviewers.

All reviewers are concerned about the lack of novelty. We understand this criticism, but we stress that our paper is not intended to introduce a new concept to the machine learning community, rather on the application of machine learning in the sciences. The call for papers clearly encourages submissions for a section “Machine Learning for Sciences”, and we believe our submission is most appropriate for this category. The novelty and contribution of our paper is the following: we have introduced an algorithm that naturally combines both physics and machine learning, without access to ground truth data. We believe this is an important contribution since a lack of trustworthy ground truth data is an issue in many inverse problems. Under these circumstances, one can only pre-train networks using synthetic data that should resemble real data, which we know is seldomly the case. Instead, we propose a method that trains on-the-fly in a self-supervised manner. One reviewer has pointed out that similar work exists and provided the reference [1]. We are aware of similar work along these lines, but we would like to stress that our network is fundamentally different in that it uses a blind-spot architecture: the provided reference is a Noise2Noise type approach which requires multiple samples with the same ‘signal’ but different noise. Such a requirement is unfeasible in many scientific applications, including for seismic deblending. Our contribution is a Noise2Void type approach. We have adapted the network introduced by Laine et al. to deal with trace-wise noise, as opposed to their target of random noise. Whilst [47] does something similar, we believe that their suggestion of using 90 and 270 degrees rotation is sub-optimal as the time axis of the two ‘views’ of the dataset are reversed and some useful correlation is lost. We instead suggest replacing rotations with flipping along the axis where noise is correlated. Whilst this is a minor difference, we believe that when it comes to differentiating a good idea to something that really works and beats SOTA, the devil always lies in the details.

All reviewers have criticized our manuscript for having a too long introduction and a too short methodology section, specifically regarding the derivation of the PnP methodology. Upon review we agree with this point. We therefore propose to merge the derivation of the ADMM in the appendix with the “method” section and explain the rationale behind incorporating the denoiser more clearly. Accordingly, we will shorten the introduction. A new version of the manuscript will be uploaded prior to the Aug 9 deadline.

All reviewers have noted that we carry out too few experiments, and one of the reviewers commented that we should have compared with more SOTA methods. There is a huge difference between the geophysical and machine learning community when it comes to open data and open-source codes.

Choice of data: given the 9-page limit, we preferred to provide insights into the proposed algorithm with a detailed ablation study instead of providing two shallow examples with different datasets. We have used Mobil AVO for two main reasons; first, using a field dataset over a synthetic dataset ensures that the complexity of the seismic wavefield is realistic. By doing so, we ensure that the results are transferable to other data (similar quality of results is expected when applying our method to other data). Second, the fact that the acquisition was not performed in simultaneous shooting mode, allows us to synthetically create the blended data and gives us the true deblended data to compare against. We believe this is the most ideal scenario to test deblending algorithms on. We have also tested with the same dataset but a different acquisition geometry, which is equally important in our field. Moreover, since our method is not pre-trained on a certain set of training data, but rather the dataset itself, we feel that the common ML pitfall of a failure to generalize to other data sets is less of an issue.

SOTA methods: whilst we would like to, unfortunately we feel that it is not possible to test alternative SOTA solvers in addition to the one already included in this work. Codes in our community are not open-source, and since methods are sometimes introduced by companies who want to keep their competitive advantage, they do not provide enough details to reproduce the experiments. The patched Fourier method is the most widely used deblending strategy in our industry, and we therefore deem it appropriate to choose this one as a comparison (code it ourselves and open-source it to the community – we cannot point to our code to avoid breaching the double-blind policy). Implementing other SOTA algorithms is a lot of extra work, which we believe would add little extra value.

---

### Meta-Review · Area_Chair_8yt4 · 2022-08-31

**Recommendation:** Reject
**Confidence:** Less certain

**Metareview:**

The paper studies a seismic deblending problem. This is a problem in reflection seismology, in which multiple excitations are applied simultaneously, and then an underdetermined inverse problem is solved to recover the underlying composition of the earth. Existing approaches to this problem are mostly based on regularization — e.g., frequency domain sparsity. The paper proposes an alternative method based on plug-and-play-ADMM, with a self-supervised regularizer. The regularizer here is a “blind spot” network, which tries to predict a pixel based on its surroundings. In simulation studies (based on synthetic blending of real seismic data), the proposed algorithm outperforms the regularization approach.

 Reviews of the paper were mixed: reviewers all recognized careful, pedagogical manner in which the paper lays out its problem of interest. At the same time, several reviewers raised concerns that the exposition was overly focused on background material, at the expense of explaining the paper’s technical contributions. Exposition aside, much of the discussion in the reviews and authors’ response centers on the novelty and depth of the paper’s technical contributions. The reviewers note that the application of self-supervised denoising within a plug-n-play framework is not a novelty of the paper (nor is it argued as one). Rather the technical contribution lies in a combination of existing ideas (self-supervised denoising ala struct BS, plug-n-play) which is well suited to the reflection seismology application. Reviewers generally felt that the paper would be stronger if it focused more on this methodology and on the technical justification of the approach. While the paper introduces a method that has value for reflection seismology, it is current form, the concerns are significant enough to place it below the bar for acceptance.

**Award:**

No

---

### Decision · Program_Chairs · 2022-09-14

Reject